# Association between Body Mass Index with Sugar-Sweetened and Dairy Beverages Consumption in Children from the Mexico–USA Border

**DOI:** 10.3390/ijerph19116403

**Published:** 2022-05-25

**Authors:** Luis Mario Gómez-Miranda, Ricardo Ángel Briones-Villalba, Melinna Ortiz-Ortiz, Jorge Alberto Aburto-Corona, Diego A. Bonilla, Pilar Pozos-Parra, Roberto Espinoza-Gutiérrez, Juan José Calleja-Núñez, José Moncada-Jiménez, Marco Antonio Hernández-Lepe

**Affiliations:** 1Sports School, Autonomous University of Baja California, Tijuana 22390, Mexico; lgomez8@uabc.edu.mx (L.M.G.-M.); briones.ricardo@uabc.edu.mx (R.Á.B.-V.); melinna.ortiz@uabc.edu.mx (M.O.-O.); jorge.aburto@uabc.edu.mx (J.A.A.-C.); espinoza.roberto@uabc.edu.mx (R.E.-G.); juan.calleja@uabc.edu.mx (J.J.C.-N.); 2Research Division, Dynamical Business & Science Society—DBSS International SAS, Bogota 110311, Colombia; dabonilla@dbss.pro; 3Research Group in Biochemistry and Molecular Biology, Universidad Distrital Francisco José de Caldas, Bogota 110311, Colombia; 4Research Group in Physical Activity, Sports and Health Sciences (GICAFS), Universidad de Córdoba, Monteria 230002, Colombia; 5Sport Genomics Research Group, Department of Genetics, Physical Anthropology and Animal Physiology, Faculty of Science and Technology, University of the Basque Country (UPV/EHU), 48940 Leioa, Spain; 6Medical and Psychology School, Autonomous University of Baja California, Tijuana 22390, Mexico; maria.pozos@uabc.edu.mx; 7Human Movement Sciences Research Center, University of Costa Rica, San Jose 11501, Costa Rica; jose.moncada@ucr.ac.cr

**Keywords:** sugar-sweetened beverages, body composition, nutrition

## Abstract

The consumption of sugar-sweetened beverages has been associated with the onset of cardiometabolic diseases. The aim of this study was to describe consumption patterns of sugar-sweetened and dairy beverages and to evaluate their correlation with the body mass index in children residing at the Mexico–USA border. A total of 722 (370 girls, 352 boys) elementary school children aged 9 to 12 years from Tijuana, Mexico, participated in the study. Anthropometric measures were recorded, and a beverage intake questionnaire was completed by the children’s parents. Significant age by sex interactions were found on body mass index Z-scores (*p* < 0.01). Boys showed higher sugar intake (*p* < 0.05) and total relative energy consumption from sugar (*p* < 0.05) than girls. The energy consumption from sugar-sweetened and dairy beverages was similar between sexes (*p* > 0.05). Sugar intake from beverages was higher than the limit recommended by the World Health Organization in boys (66%) and girls (44%). A high frequency of consumption of sugar-sweetened beverages and similar intake of dairy beverages were found in children from the Mexico–USA border. The high consumption of sugar exceeds international recommendations and should be carefully monitored.

## 1. Introduction

For most people in the world, nutritional behaviors have changed as a direct consequence of the increase in food availability, a dramatic change in portion size and aggressive marketing strategies [1]. The intake of hypercaloric meals, foods with poor vitamin and mineral content, foods high in fat, salt, and sugar, and high-calorie beverages have significantly increased in modern industrialized society [2]. This extra caloric consumption along with many other biological, social and/or environmental determinants has contributed substantially to the development of overweight and obesity, thereby impacting public health worldwide [3,4]. In addition, childhood overweight and obesity are closely related to the development of other chronic non-communicable diseases such as type 2 diabetes mellitus, metabolic syndrome, certain types of cancer, and arterial hypertension earlier in life [5,6].

The dietary consumption of sugars in children and adolescents has dramatically exploded, particularly from sugar-bakery products, dairy products, and sugar-sweetened beverages (SSB) [7]. These beverages have high glycemic and low satiety indexes, which causes a higher food consumption following their intake [8]. Evidence from systematic reviews and meta-analyses indicates that SSB (e.g., juices and popular carbonated drinks) increase body mass index (BMI), causes gut dysbiosis, increase the risk of inflammatory bowel disease and, as a whole, a higher predisposition towards cardio-metabolic diseases in children and adults [9,10,11,12].

Research performed in Latin American countries has shown that the consumption of beverages with high energy content represents an important risk factor for the development of overweight, obesity, and other non-communicable diseases [13]. Different studies reported an increase in the number of school-aged children suffering from overweight and obesity because of an increased intake of SSB [14]. In the Hispanic race, the consumption of these drinks and other high-energy foods in children is associated with television advertisements [15]. The marketing strategies used by the large soft drink companies stimulate the consumption of their products since they use specific population-targeted messages in different media [16]. Nonetheless, overweight and obesity figures in North American children and adolescents have increased in the last 25 years [17]. To counteract this situation, the World Health Organization has recommended engaging in regular physical activity, increasing consumption of fruits and vegetables and limiting the intake of total fats and sugars [18].

Nowadays, sugars from beverages are the main source of added sugars, contributing to 10% of the overall energy intake, which is why different political strategies have been established to discourage their consumption [19]. In January 2014, a tax on SSB was enforced in Mexico with the main goal of reducing caloric and sugar intake for obesity prevention, but the effects in children are still unknown. Additionally, a clear understanding of the nutrient profile of beverages consumed in children to determine if the SSB tax is having a positive effect is needed.

There is scientific evidence suggesting that dairy products, particularly 1% skim milk, lower the risk of obesity in children and adolescents [20]. This type of beverage negatively modulates caloric intake, increases satiety compared to SSB, and therefore reduces repetitive consumption resulting in a lower total daily energy intake. However, a meta-analysis [21] showed that the protective effect of dairy products might be modestly seen in adolescents, and it is inconsistently found during early and middle childhood [22]. This behavior is largely explained by the decrease in milk consumption, since only a third of adolescents have reported regular consumption (4–6 times per week). In addition, the purported association between milk intake and reduced obesity has not been confirmed in different populations [23]; therefore, culturally sensitive studies are needed. Thus, the purpose of the study was to determine the correlation of the consumption of SSB and dairy beverages with the BMI Z-score in children from the Mexico–USA border.

## 2. Materials and Methods

This cross-sectional, descriptive-analytical study was conducted following the STrengthening the Reporting of OBservational studies in Epidemiology (STROBE) statement for reports of cross-sectional studies recommendations.

### 2.1. Eligibility Criteria

Seven hundred and twenty children (352 boys, 370 girls) were recruited from seven elementary schools in the School Zone N° 25 of the city of Tijuana from the Mexico-USA border. Children were allowed to participate if they met the following inclusion criteria: (a) children over 9 years old and under 13 years old, (b) officially registered in the participating schools, (c) did not report metabolic diseases by the parents in the interview, and (d) were registered in fourth to sixth grade of elementary-middle school.

### 2.2. Socio-Demographics

The socio-demographic characteristics (age, sex, and race/ethnicity) were collected from the participating child and the head of household during a personal interview with the child’s parents. Additionally, marital status, education level, and income were collected about the head of household. The ratio of income to poverty was estimated using family income to the poverty level in the North of Mexico, using adjustments for family size and residence colony.

### 2.3. Procedures

The anthropometric assessments of children were conducted in private rooms within each of the schools, and parents completed the beverage intake questionnaire in regular classrooms. Children’s body height was measured with a portable stadiometer (Seca Model 214, Seca Corp., Hanover, MD, USA) recording the nearest 0.1 cm. The body mass was measured with an electronic scale to the nearest 0.1 kg (Tanita Corp., Tokyo, Japan), having the participant barefoot with light clothes and standing still on top of the scale. The BMI was calculated by taking the participants body mass, in kilograms, divided by their stature in meters squared (BMI = body mass (kg)/[stature (m)]^2^). The BMI was the base to calculate the age and sex specific BMI Z-scores, classifying as a normal body mass a value under +1 standard deviation (SD) and as excess of body mass a value above +1 SD (above +1 SD are considered as overweight, and above +2 SD as obese), according to the World Health Organization Growth Reference median [24]. For each child, BMI Z-score was calculated using age in months with WHO AnthroPlus software [25].

The Beverage Intake Questionnaire (Appendix A), was used to determine fluid intake. The instrument lists several drinks (water, 100% fruit juice, sweetened juice beverages, whole milk, low-fat milk, soft drinks, to mention some) ingested daily and parents were asked to indicate the frequency and quantity consumed by their children with the help of portion size charts [26]. Then, the caloric content calculation of the beverages was determined by the nutritional labels or the Mexican System of Equivalent Foods [27].

### 2.4. Outcome Measures

The primary outcome was the BMI Z-score of children from the Mexico–USA border. The daily intake of: total energy, sugar, SSB, dairy beverages, and sugar-free beverages were reported as secondary outcomes.

### 2.5. Statistical Methods

Statistical analyses were performed with the IBM-SPSS Statistics, version 23 (SPSS Inc., IBM Company, Chicago, IL, USA) (Appendix A). Data distribution normality was examined by the Shapiro–Wilk test, and the homoscedasticity by the Levene test, for each group. Descriptive statistics are presented as mean and standard deviation (SD), or, when specified, standard error of the mean (SEM). Statistical differences between children sex or body mass (healthy and body mass excess) were compared using paired t-tests. To evaluate the association among variables, a Pearson correlation was performed by sex at the different age categories between BMI Z-score with energy, sugar intake, and energy from sugar-sweetened or dairy beverages consumption. The level of significance was set a priori at *p* < 0.05.

### 2.6. Ethics

The Institutional Review Board of the Autonomous University of Baja California approved the protocol (Ethical Approval Code: UABC/2018.02-E01) (Appendix A). The purpose of the study was explained to the principal of each school and upon approval, the parents or legal guardians were contacted. Parents allowing their children to participate in the study read and signed an informed consent where the methodology, potential risks and benefits of the study were thoroughly explained.

## 3. Results

All children (*n* = 722) and their parents were of the Hispanic ethnic group (51% were females). For both boys and girls, 49% were at excess body mass (BMI Z-score above +1 SD), being the highest percentage the girls of 12 years (75%) (Table 1). The head of household mean age was 38 ± 2 years, with 69% of income to poverty considered low in Tijuana. Only 43% reported education of at least high school, and 69% reported being partnered. Anthropometric characteristics showed BMI Z-scores were different between age groups for both sexes (*p* < 0.01).

As previously reported [23], the internal consistency of the Beverage Intake Questionnaire responses was acceptable (Cronbach’s α = 0.73). The daily beverage consumption by sex is presented in Table 2. In general, boys reported (*p* ≤ 0.05) a higher sugar (g/day) and SSB (kcal/day) consumption than girls (53.9 ± 3 vs. 44.1 ± 3 and 229.1 ± 14 vs. 191.2 ± 13, respectively). The daily energy and dairy beverage consumption were not different between boys and girls, regardless of whether they were of normal weight or excess body mass (*p* < 0.05).

The daily sugar consumption from beverages by sex and age clusters is depicted in Figure 1. A high consumption of sugars (>25 g) was positively associated according to age groups. Additionally, boys reported a higher total relative energy intake (%) from sugar than girls (57.2 ± 25 vs. 52.2 ± 26; *p* = 0.008).

Interestingly, BMI Z-score in girls and boys did not present significant correlations with SSB intake (Table 3), only girls with excess body mass showed a positive correlation with sugar-free beverages consumption. In contrast, dairy and sugar-free beverages intake in the total of boys and boys with excess body mass were negatively correlated with the BMI Z-score.

## 4. Discussion

In this study, we determined the association of BMI Z-score with the consumption of sugar-sweetened and dairy beverages in children from the Mexico–USA border city of Tijuana, Mexico. The relevance of the topic lies in that the frequent consumption of SSB has been associated with a plethora of cardiometabolic diseases, and milk consumption has been inversely related to children obesity [28,29,30].

We found that boys consumed more SSB than girls (66% vs. 44%). This sex difference as been also reported in Nanjing, China, on a large sample of 9 to 12 years old children (N = 5193), where boys consumed more SSB than girls (52.9% vs. 43.6%) [31]. This behavior has been explained previously, and the strongest theories propose that girls tend to be more concerned about their body aesthetics and take actions to correct their weight, such as including more fruits and vegetables in their diet, exercising more or consuming fewer sugar-based drinks. Ultimately, this in turn contributes to an increased risk of eating disorders [32]. The reported sugar consumption in both countries is higher than the recommended by international organizations [33]. For instance, in the United States, the Department of Health and Human Services and the Department of Agriculture recommend limiting the consumption of added sugars to less than 10% of calories per day [34].

Recent evidence suggests that the sugar intake recommendations should be questioned since these are based on low to very-low quality scientific evidence, a claim that raised a heated debate among scholars with opposing points of view on the subject [35,36]. For instance, evidence from systematic reviews showed an association between sugar intake from beverages and obesity, type 2 diabetes mellitus, hypertension and cardiovascular risk in children and adults [37,38,39]; however, others are more precautious and suggest long-term clinical studies aiming at understanding the association between sugar intake consumption and the onset of cardiometabolic diseases [40].

The elevated children and adolescent obesity are mostly influenced by behavioral patterns like sugar consumption worldwide [41]. For instance, in undernourished and obese 6- to 12-years old South African children, the sugar intake ranged from 7.2% to 11.7% of daily calories [42]. In a representative sample of the Spanish population aged 9 to 75 years old, the median total sugar intake was 17%; this is, 7% higher than the current recommendations, however, SSB like soft drinks only represented a small percentage (2.24%) of the total sugar intake, where surprisingly, dairy products were the biggest contributors (23.2%) [34,43].

As mentioned above, we found that girls have a lower SSB, but a higher energy consumption from beverages than boys. This finding might be explained by girls reporting a higher consumption of milk-based beverages, but less SSB than boys. Girls in general tend to take more care of their health and body aesthetics, they are more concerned about implementing weight control strategies and are more receptive to peer influence and advertising [44]. This type of beverages has been promoted as a protective factor against the development of obesity and other diseases, considering dairy beverages are a nutrient-rich source of protein, calcium, riboflavin, and vitamins, and recent research provides evidence that milk and other dairy products intake are inversely associated with obesity in children [29,45]. In a research conducted in Spain, girls with normal body mass had a higher consumption of dairy beverages compared to girls with overweight or obesity [19].

The WHO recommends an intake of free sugars roughly 25 g (below 5% of total energy intake) in an adult with normal BMI [46]. Our results suggest that most children between 9 and 12 years consume more sugar from beverages than the WHO guidelines recommendations in adults, which is alarming due the 5% of total energy intake in children is still fewer 25 g/day, also, sugar intake of solid foods are not considered in this research. One of the reasons explaining the elevated consumption of SSB in children and adolescents is the phenomenon of ‘nutritional transition’, which is characterized by the consumption of processed foods with high added amounts of trans fat, sugar and salt. These foods are usually ingested as fast-food meals and for most people, have accessible prices. This nutritional pattern of high-energy meal consumption has been described in Mexico before, especially in populations from the northern border of the country [47].

Another potential variable related to the increased sugar intake may be the marketing strategies used by the beverage industry, since they use targeted messages aimed at motivating the consumption of their products, there are studies indicating an association between SSB consumption and the wide range of commercial advertisements on television during children’s schedules [16,19]. Therefore, parents and/or legal guardians must control media exposition by children and food consumption related to sedentary time since ‘metabolic programming’ leads to several diseases (e.g., obesity, hypertension) starts during childhood [48]. Finally, low schooling and/or acculturation has impacted in the economic, social, and cultural processes of Mexico, particularly in low socioeconomic strata. These phenomena have been previously described and expose the influence that they exert on the absence of informed decisions when consuming healthy foods and how they are replaced by affordable foods or beverages, with high availability in the local market but low nutritional value [49].

In this study, we found higher consumption of sugar from beverages in boys than in girls; a finding consistent with results from the intake trend around 15 countries reported in an actual systematic review [50]. In addition, Della Corte et al. (2021), concluded that North American children and adolescents’ consumption of sugar and SSB has increased substantially in the last two decades, matching the results found in this study that are significantly higher than those reported in a sample of children from 5 to 9 years old in South America [51]. Therefore, our findings on the high sugar intake might cause elevated BMI Z-scores and could be reasonably considered a potential trigger for cardiovascular and metabolic diseases in children from the Mexico–USA border sample studied.

The principal limitation of this study is the lack of physical activity information in the childhood, moreover, public schools in Mexico offer only one hour of physical education lessons per week, with an effective time of 39 ± 0.7 min, of which only 41% (15 min) is intended for moderate-to-vigorous intensity activities such as soccer or basketball. In this sense, physical activity in elementary school does not correspond to the minimum recommendations for a healthy life, and additional strategies are required for the promotion and achievement of well-being [52,53]. Other limitations of the present study are related to the implemented questionnaire, which, even though it has been validated in different populations, has been shown to overestimate daily intake of beverage or foods due the self-reported bias and the lack of information obtained about the daily sugar intake from solid foods [8]. Finally, there is a limitation to determine causality between BMI and the factors evaluated due we have carried out only a cross-sectional study, but it is important to emphasize that this can be a starting point to design longitudinal studies focused on determining the causes and possible nutritional and physical activity programs as strategies that seek to reduce this health problem.

According to the aforementioned, there is an urgency to develop strategies focused to counteract the high rate of childhood overweight and obesity. Future strategies should consider a multidimensional model that includes, among other determinants: healthy food options in school stores, developing food preparation skills, encouraging physical activity beyond school (by offering different types of exercises or sports to increase the possibility that children will be motivated to move by finding something they like) and consider to evaluate more specific parameters of body composition, such as waist girth, hip girth, and body fat percentage by bioelectrical impedance analysis, anthropometry or densitometric methods. We consider that the adaptation of protocols that contemplate a holistic view of this problem and the consideration of specific markers, would result in more effective strategies that could help them adopt a healthy lifestyle that lasts throughout life.

## 5. Conclusions

An elevated percentage of children showed higher sugar consumption from beverages than those amounts recommended by the World Health Organization. In general, boys consumed more added sugar than girls; however, boys consumed less sugar-free beverages, which is correlated with a higher BMI Z-score. There were no differences in the consumption of dairy beverages between boys and girls. The present study strengthens the evidence that obesity spreads affecting from an early age. Then, the efforts to reverse obesity should focus on early childhood care.

## Figures and Tables

**Figure 1 ijerph-19-06403-f001:**
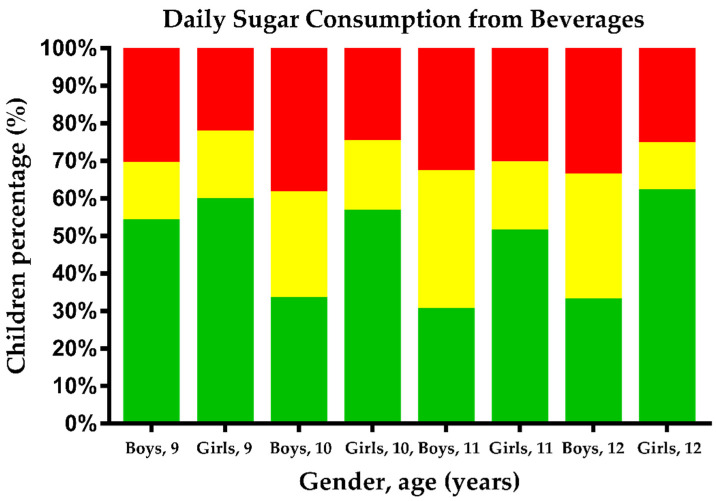
Daily sugar consumption from beverages in children by gender. Bars in green represent the percentage of children who consume less than 25 g of sugar per day, bars in yellow represent those who consume from 25 to 50 g of sugar per day, and bars in red represent those who consume more than 50 g of sugar per day.

**Table 1 ijerph-19-06403-t001:** Anthropometric characteristics in children from the Mexico–USA border.

Variables	Age Group (Years)			
	9 (*n* = 33)	10 (*n* = 176)	11 (*n* = 117)	12 (*n* = 26)
Boys (*n* = 352)				
Stature (cm)	136.2 ± 0.01	140.4 ± 0.01	146.5 ± 0.01	150.9 ± 0.02
Body mass (kg)	39.7 ± 2.3	41.2 ± 0.9	42.9 ± 0.9	45.1 ± 2.4
BMI Z-score	1.23 ± 0.2	0.94 ± 0.1	0.64 ± 0.1	0.17 ± 0.3
Normal body mass (*n*)	13 (39%)	76 (43%)	65 (56%)	16 (62%)
Excess of body mass (*n*)	20 (61%)	100 (57%)	52 (44%)	10 (38%)
	9 (*n* = 50)	10 (*n* = 186)	11 (*n* = 118)	12 (*n* = 16)
Girls (*n* = 370)				
Stature (cm)	134.3 ± 0.01	140.5 ± 0.01	147.3 ± 0.01	152.1 ± 0.02
Body mass (kg)	34.4 ± 1.4	39.7 ± 0.7	44.3 ± 0.9	58.4 ± 4.8
BMI Z-score	0.58 ± 0.2	0.73 ± 0.1	0.64 ± 0.1	1.32 ± 0.2
Normal body mass (*n*)	29 (48%)	94 (43%)	66 (56%)	4 (25%)
Excess of body mass (*n*)	21 (42%)	92 (57%)	52 (44%)	12 (75%)

Data presented as mean ± SEM. BMI: Body mass index.

**Table 2 ijerph-19-06403-t002:** Beverage intake pattern by children from the Mexico–USA border.

Variables	Age Group (Years)				BMI Z-Score	
Boys	9	10	11	12	NBM	EBM
Sugar intake (g/day)	45.4 ± 8	52.3 ± 4	57.9 ± 7	57.6 ± 12	54.9 ± 4	53.0 ± 4
Energy (kcal/day)	281.3 ± 44	381.0 ± 24	4342.3 ± 39	396.2 ± 61	417.4 ± 28	371.2 ± 25
SS-B (kcal/day)	189.1 ± 35	221.5 ± 16	246.7 ± 30	253.9 ± 55	229.4 ± 19	229.0 ± 20
DB (kcal/day)	77.1 ± 14	142.4 ± 14	177.8 ± 21	169.8 ± 30	173.2 ± 17	128.7 ± 12
SF-B (mL/day)	593.8 ± 100	710.1 ± 52	836.4 ± 59	808.2 ± 142	873.1 ± 57	633.8 ± 43
Water (mL/day)	529.1 ± 100	562.8 ± 55	714.0 ± 73	654.6 ± 163	707.0 ± 61	522.6 ± 48
SS-B (mL/day)	534.1 ± 89	654.7 ± 41	696.2 ± 73	607.0 ± 107	671.3 ± 46	638.0 ± 49
DB (mL/day)	129.6 ± 22	240.4 ± 24	296.6 ± 36	280.0 ± 48	293.3 ± 29	213.3 ± 19
**Girls**	**9**	**10**	**11**	**12**	**NBM**	**EBM**
Sugar intake (g/day)	38.8 ± 8	42.5 ± 4	49.8 ± 6	39.3 ± 11	43.7 ± 4	44.6 ± 4
Energy (kcal/day)	318.3 ± 63	368.0 ± 33	410.8 ± 44	291.0 ± 68	363.3 ± 33	380.3 ± 33
SS-B (kcal/day)	174.3 ± 40	182.0 ± 18	213.2 ± 25	191.7 ± 62	191.8 ± 19	190.6 ± 19
DB (kcal/day)	129.8 ± 29	173.8 ± 19	185.1 ± 25	89.4 ± 12	159.0 ± 17	177.1 ± 20
SF-B (mL/day)	728.5 ± 108	751.3 ± 48	855.2 ± 60	640.7 ± 144	756.3 ± 48	798.0 ± 50
Water (mL/day)	523.1 ± 82	637.4 ± 57	739.0 ± 76	623.7 ± 213	626.1 ± 53	665.1 ± 60
SS-B (mL/day)	515.3 ± 100	516.7 ± 47	587.0 ± 67	573.8 ± 186	518.3 ± 49	566.4 ± 51
DB (mL/day)	225.4 ± 50	295.6 ± 33	304.2 ± 40	146.9 ± 19	267.1 ± 28	298.7 ± 35

Data presented as mean ± SEM. BMI: Body Mass Index; NBM = Normal body mass; EBM = Excess body mass; SS-B: SSugar-sweetened beverages; DB: Dairy beverages; SF-B: Sugar-free beverages.

**Table 3 ijerph-19-06403-t003:** Correlations between BMI Z-score with sugar intake, energy, sugar-sweetened, dairy, and sugar-free beverages.

Population/Subgroup	Sugar Intake (g/day)	Energy (kcal/day)	SS-B (kcal/day)	DB (mL/day)	SF-B (kcal/day)
Total of children	0.028	−0.030	−0.032	−0.035	−0.070
Girls (Total)	−0.002	0.011	−0.017	0.025	0.056
Girls (NBM)	−0.080	−0.049	−0.091	0.025	0.005
Girls (EBM)	0.106	0.068	0.093	0.007	0.155 *
Boys (Total)	−0.062	−0.083	−0.054	−0.105 *	−0.185 **
Boys (NBM)	−0.072	−0.023	−0.088	0.072	0.008
Boys (EBM)	−0.126	−0.195 **	−0.121	−0.210 **	−0.239 **

BMI: Body Mass Index; NBM: Normal body mass; EBM = Excess body mass; SS-B: Sweetened beverages; DB: Dairy beverages; SF-B: Sugar-free beverages. Asterisk (*) means *p* < 0.05, Double Asterisk (**) means *p* < 0.01.

## Data Availability

Any researcher that contacts this project Principal Investigator, L.M.G.-M. (lgomez8@uabc.edu.mx) will have access to the study data, in accordance with the informed consent provided by participants on the use of confidential data.

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
