# Peer review of "Association between Body Mass Index with Sugar-Sweetened and Dairy Beverages Consumption in Children from the Mexico–USA Border"

_ijerph, 2022, doi:10.3390/ijerph19116403_

Round 1

Reviewer 1 Report

This is an interesting relationship to explore. However there are some major issues to address before this manuscript is ready for publication.

General:

For review purposes, it would be useful to provide line numbers.

The article needs to be proofread as the grammatical errors make it difficult to read. In the introduction section, some examples are given, along with suggestions on how to improve it. Please continue to edit the whole manuscript.

Introduction

As the determinants of overweight and obesity are multifactorial, please reword sentence to Positive energy balance is one of the contributors to obesity…..(approx. line 5 of intro).

Are there any research using measures other than BMI to study associations between weight and SSB consumption?

The authors state that milk protects children as it has the opposite satiating effect to SSB but please clarify if reference 18 states that the finding was specific to plain and or low fat milk? Ref 19 and 20- I bit more background needed here- what were some of the inconsistent findings? As this is the reason which leads to your analysis, there needs to be more detail here.

The sentence starting on line 6 would better if written:

“In addition, childhood overweight and obesity are closely related to the development of other chronic non-communicable diseases such as type 2 diabetes mellitus, metabolic syndrome, certain types of cancer, and arterial hypertension, earlier in life [5,6].

This sentence needs revising: There exists a children’s excessive sugar intake worldwide

In this sentence, it should be “reported”: Different studies report an increase in the number of school-aged children suffering from overweight and obesity because of an increased intake of SSB [12].

Spelling: In te Hispanic race,

This sentence does not read clearly. Remove the being and start the sentence with SSP: In Mexico, overweight and obesity is present in 33% of children and 36% of adolescents, being SSB the main source of added sugars, contributing with 10% of the overall energy intake [17].

Materials and Methods

Spelling: Mexico-Us should be US or USA

Inclusion criteria. Is age 9.00-12.99 or 9.00-11:99? There is almost 12 months difference between the two and this difference is significant in terms of growth. Please clarify.

Please provide justification why the WHO cut-offs for BMI for age were used instead of CDC. This is not wrong, but with so many available cut-offs, a justification is warranted- for example this might be the reference used in clinical practice in Mexico

Please check the WHO BMI cut-offs that you have used. The WHO does not classify obesity as +3SD for your age group which falls in the 5-19 years category. Please check the reference I have provided below for WHO BMI for age cut-offs. If using the WHO BMI for age cut-offs, it would be better to reference them, and not the paper you have referenced (ref 22).

https://www.who.int/news-room/fact-sheets/detail/obesity-and-overweight

Children under 5 years of age

For children under 5 years of age:

overweight is weight-for-height greater than 2 standard deviations above WHO Child Growth Standards median; and

  • obesity is weight-for-height greater than 3 standard deviations above the WHO Child Growth Standards median.
  • Charts and tables: WHO child growth standards for children aged under 5 years

Children aged between 5–19 years

Overweight and obesity are defined as follows for children aged between 5–19 years:

  • overweight is BMI-for-age greater than 1 standard deviation above the WHO Growth Reference median; and
  • obesity is greater than 2 standard deviations above the WHO Growth Reference median.
  • Charts and tables: WHO growth reference for children aged between 5–19 years

Additionally, the paper you have referenced (ref 22) actually says also says this: “For older children, the WHO adolescence BMI-for-age curves at 19 years closely coincide with adult overweight (BMI 25) at +1 SD and adult obesity (BMI 30) at +2 SD. As a result, these SD classifications are extended down to 5 years”. The paper is an opinion piece and should not be used as the main reference for your choice to use the selected cut-offs.

z-score should be Z-score

Was the WHO anthro software (AnthroCalc) used to calculate these Z-scores? If this software was used, then the program would have automatically calculated the correct Z-scores based on the info the authors provided, and it is only the manuscript that needs editing. If this is the case please edit the manuscript and provide reference to AnthroCalc.

if AnthroCalc was NOT used then all of the data would need to be reanalysed using the correct cut-offs for overweight and obesity. Please update results accordingly.

Results:

Based on the point made above, there is no need for me to review any of the results involving weight status, until the authors confirm how exactly the Z-scores were calculated and reanalyse where necessary.

Overall though, the results are only descriptive. There is hardly any actual statistical analysis. For example, in table 3, why only correlations? Why not some regression models? The use of Z-scores will already account for age and sex so you can analyse the sample as a whole and adjust for other demographic confounders? And then highlight differences you note between sexes and age groups?

The results overall are quite hard to follow. The lack of sequence and grammatical issues are what contributes to this. For example in the middle of explaining Figure 1, he authors start talking about energy from dairy when the figure is specifically for daily sugar consumption. The paragraph starting with “In correlation analysis….” is difficult to understand. Could some of the negative correlations actually mean that underreporting could be an issue?

Discussion:

Why are we seeing that boys consume more soft drinks to girls?

Why are SSBs contributing to a significant proportion of energy intake? The authors fail to provide some insight into this? are we seeing these trends in lower SES where it is cheaper to access these foods? Are advertising tarting lower SES?

Why are girls consuming more milk? In your introduction, you mention advertising as something to do with these trends. Are advertisements for dairy targeting females or proposed benefits for females?

Overall, the discussion needs much more explanation of the trends rather than just stating it is consistent with literature. A more in depth investigation into the “why” means that there are future research avenues to pursue in order to see change

Author Response

Thank you for the time devoted to our article entitled: “Association Between Body Mass Index with Sugar-Sweetened and Dairy Beverages Consumption in Children from the Mexico-USA Border”, authored by Luis Mario Gómez-Miranda, Ricardo Ángel Briones-Villalba, Melinna Ortiz-Ortiz, Jorge Alberto Aburto-Corona, Diego A Bonilla, Pilar Pozos-Parra, Roberto Espinoza-Gutiérrez, Juan José Calleja Nuñez, José Moncada-Jiménez, Marco Antonio Hernández-Lepe (corresponding author). We addressed all the comments in a new version of our manuscript (ijerph-1687756-R1) according to your suggestions and comments. Revisions and modifications were highlighted in red within the document and a point-by-point response is described below.

This is an interesting relationship to explore. However there are some major issues to address before this manuscript is ready for publication.

 General:

 For review purposes, it would be useful to provide line numbers.

 R= Thank you. The line numbers have been indicated for your convenience.

The article needs to be proofread as the grammatical errors make it difficult to read. In the introduction section, some examples are given, along with suggestions on how to improve it. Please continue to edit the whole manuscript.

R= Several authors have reviewed the manuscript in order to improve the English quality. Either way, we are in a position to send it to the MDPI English Editing Service.

Introduction

As the determinants of overweight and obesity are multifactorial, please reword sentence to Positive energy balance is one of the contributors to obesity…..(approx. line 5 of intro).

R= Thank you. The following statement has been added according to your suggestion: “This extra caloric consumption along with many other biological, social and/or environmental determinants has contributed substantially to the development of overweight and obesity, thereby impacting public health worldwide”.

Are there any research using measures other than BMI to study associations between weight and SSB consumption?

R= Thank you. The following statement has been adjusted according to your suggestion:  “Evidence from systematic reviews and meta-analyses indicates that SSB (e.g., juices and popular carbonated drinks) increase body mass index (BMI), causes gut dysbiosis, increase the risk of inflammatory bowel disease and, as whole, a higher predisposition towards cardio-metabolic diseases in children and adults”.

The authors state that milk protects children as it has the opposite satiating effect to SSB but please clarify if reference 18 states that the finding was specific to plain and or low-fat milk? Ref 19 and 20- I bit more background needed here- what were some of the inconsistent findings? As this is the reason which leads to your analysis, there needs to be more detail here.

 R= Thank you. Two statements have been added according to your suggestion: “There is scientific evidence suggesting that dairy products, particularly 1% skim milk, lower the risk of obesity in children and adolescents [20]. This type of beverage negatively modulates caloric intake, increases satiety compared to SSB, and therefore reduces repetitive consumption resulting in a lower total daily energy intake” and “This behavior is largely explained by the decrease in milk consumption, since only a third of adolescents have reported regular consumption (4-6 times per week)”.

The sentence starting on line 6 would better if written:

“In addition, childhood overweight and obesity are closely related to the development of other chronic non-communicable diseases such as type 2 diabetes mellitus, metabolic syndrome, certain types of cancer, and arterial hypertension, earlier in life [5,6].

R= Thank you. The sentence has been modified according to the suggestion.

This sentence needs revising: There exists a children’s excessive sugar intake worldwide

R= Thank you. The following statement has been adjusted according to your suggestion: “The dietary consumption of sugars in children and adolescents has dramatically exploded, particularly from sugar-bakery products, dairy products, and sugar-sweetened beverages (SSB)”.

In this sentence, it should be “reported”: Different studies report an increase in the number of school-aged children suffering from overweight and obesity because of an increased intake of SSB [12].

R= Thank you. The sentence has been modified according to the suggestion.

Spelling: In te Hispanic race,

R= Thank you. The word has been modified.

This sentence does not read clearly. Remove the being and start the sentence with SSP: In Mexico, overweight and obesity is present in 33% of children and 36% of adolescents, being SSB the main source of added sugars, contributing with 10% of the overall energy intake [17].

R= Thank you. The following statement has been adjusted according to your suggestion: “Nowadays, sugars from beverages are the main source of added sugars, contributing to 10% of the overall energy intake, which is why different political strategies have been established to discourage their consumption”.

Materials and Methods

Spelling: Mexico-Us should be US or USA

R= Thank you. The word has been modified.

Inclusion criteria. Is age 9.00-12.99 or 9.00-11:99? There is almost 12 months difference between the two and this difference is significant in terms of growth. Please clarify.

R= Thank you. The following statement has been adjusted according to your suggestion: “children over 9 years old and under 13 years old”.

Please provide justification why the WHO cut-offs for BMI for age were used instead of CDC. This is not wrong, but with so many available cut-offs, a justification is warranted- for example this might be the reference used in clinical practice in Mexico

R= Thank you. The published study by Valerio, G., Balsamo, A., Baroni, M. G., Brufani, C., Forziato, C., Grugni, G., ... & Manco, M. (2017). Childhood obesity classification systems and cardiometabolic risk factors: a comparison of the Italian, World Health Organization, and International Obesity Task Force references. Italian Journal of Pediatrics, 43(1), 1-7, determinate the significance of various cut-off points for BMI, with the WHO system being the most sensitive for determining cardiometabolic risk in the overweight or obese community. Given the seriousness of the obesity epidemic, the authors believe that the use of the WHO system should be preferred over other international standards of clinical practice and/or obesity screening.

Please check the WHO BMI cut-offs that you have used. The WHO does not classify obesity as +3SD for your age group which falls in the 5-19 years category. Please check the reference I have provided below for WHO BMI for age cut-offs. If using the WHO BMI for age cut-offs, it would be better to reference them, and not the paper you have referenced (ref 22).

https://www.who.int/news-room/fact-sheets/detail/obesity-and-overweight

Children under 5 years of age

For children under 5 years of age:

overweight is weight-for-height greater than 2 standard deviations above WHO Child Growth Standards median; and

  • obesity is weight-for-height greater than 3 standard deviations above the WHO Child Growth Standards median.
  • Charts and tables: WHO child growth standards for children aged under 5 years

Children aged between 5–19 years

Overweight and obesity are defined as follows for children aged between 5–19 years:

  • overweight is BMI-for-age greater than 1 standard deviation above the WHO Growth Reference median; and
  • obesity is greater than 2 standard deviations above the WHO Growth Reference median.
  • Charts and tables: WHO growth reference for children aged between 5–19 years

 R= Thank you for your observation, we have modified the cut offs according to the WHO and referenced the one that you provided us.

Additionally, the paper you have referenced (ref 22) actually says also says this: “For older children, the WHO adolescence BMI-for-age curves at 19 years closely coincide with adult overweight (BMI 25) at +1 SD and adult obesity (BMI 30) at +2 SD. As a result, these SD classifications are extended down to 5 years”. The paper is an opinion piece and should not be used as the main reference for your choice to use the selected cut-offs.

R= We have eliminated that reference from the manuscript, thanks for your observation.

z-score should be Z-score

R= Thank you. The expression has been modified along the document.

Was the WHO anthro software (AnthroCalc) used to calculate these Z-scores? If this software was used, then the program would have automatically calculated the correct Z-scores based on the info the authors provided, and it is only the manuscript that needs editing. If this is the case please edit the manuscript and provide reference to AnthroCalc.

if AnthroCalc was NOT used then all of the data would need to be reanalysed using the correct cut-offs for overweight and obesity. Please update results accordingly.

 R= We think you mean WHO AnthroPlus Software, and effectively, it was the software used to calculate BMI Z-scores. According to your recommendation, it is referenced now, thank you.

Results:

Based on the point made above, there is no need for me to review any of the results involving weight status, until the authors confirm how exactly the Z-scores were calculated and reanalyse where necessary.

R= Information about calculation of BMI Z-score was added in Section 2.3, including the following statement: “The BMI was the base to calculate the age and sex specific BMI Z-scores, classifying as a normal body mass a value under +1 standard deviation (SD) and as excess of body mass a value above +1 SD (above +1 SD are considered as overweight, and above +2 SD as obese), according to the World Health Organization Growth Reference median. For each child, BMI Z-score was calculated using age in months with WHO AnthroPlus software”.

The results overall are quite hard to follow. The lack of sequence and grammatical issues are what contributes to this. For example in the middle of explaining Figure 1, he authors start talking about energy from dairy when the figure is specifically for daily sugar consumption. The paragraph starting with “In correlation analysis….” is difficult to understand. Could some of the negative correlations actually mean that underreporting could be an issue?

R= Thank you. Some paragraphs have been restructured; the changes were:

“The daily sugar consumption from beverages by sex and age clusters is depicted in Figure 1. A high consumption of sugars (>26g) was positively associated according to age groups. Additionally, boys reported a higher total relative energy intake (%) from sugar than girls (57.2±25 vs. 52.2±26; p= 0.008)”.

“Interestingly, low energy intake, sugars in general, and sugar-sweetened drinks were negatively correlated with lower body weight in both genders (NBM). In contrast, dairy consumption was positively correlated with a healthy weight within the same groups. According to the analysis of subgroups by age in Figure 1, the 11-year-olds showed the highest consumption of sugars from beverages, this subgroup in turn correlates positively with body mass gain according to the BMI Z-score”.

Discussion:

Why are we seeing that boys consume more soft drinks to girls?

R= Several studies indicate that girls are more concerned with their body shape and tend to eat healthier (i.e., more fruit, vegetables, and less SSB) relative to boys, even to suffer disorders in eating behavior. This argument has been included in the discussion section. Thank you.

Why are SSBs contributing to a significant proportion of energy intake? The authors fail to provide some insight into this? are we seeing these trends in lower SES where it is cheaper to access these foods? Are advertising tarting lower SES?

R= The next paragraph has been added: “Finally, low schooling and/or acculturation has impacted in the economic, social, and cultural processes of Mexico, particularly in low socioeconomic strata. These phenomena have been previously described and expose the influence that they exert on the absence of informed decisions when consuming healthy foods and how they are replaced by affordable foods or beverages, with high availability in the local market but low nutritional value”. Thanks for your observation.

Why are girls consuming more milk? In your introduction, you mention advertising as something to do with these trends. Are advertisements for dairy targeting females or proposed benefits for females?

R= Women in general tend to take more care of their health and body aesthetics, they are more concerned about implementing body mass control strategies and are more receptive to advertising and peer influence. This type of beverages has been promoted as a protective factor against the development of obesity and other diseases, this due dairy beverages are nutrient rich and a source of protein, calcium, riboflavin, and vitamins, and recent research provides evidence that milk and other dairy products intake are inversely associated with obesity in children. This argument has been added to the discussion. Thank you.

Overall, the discussion needs much more explanation of the trends rather than just stating it is consistent with literature. A more in depth investigation into the “why” means that there are future research avenues to pursue in order to see change

R= We appreciate your time and support. We hope that attention to the points that have been discussed in all the manuscript sections culminated in a more important research work.

Reviewer 2 Report

The indications made in the first round have been solved, although the general appreciations are maintained.

Perhaps the conclusions section could be slightly expanded, but much of the content for this section is already laid out in the Discussion.

Author Response

The indications made in the first round have been solved, although the general appreciations are maintained.

Perhaps the conclusions section could be slightly expanded, but much of the content for this section is already laid out in the Discussion.

R= We hope that attention to the points that have been discussed in all the manuscript sections culminated in a more important research work. We appreciate your time and support to improve the present manuscript.

Reviewer 3 Report

The manuscript have been corrected and can be accepted as present form.

Author Response

The manuscript have been corrected and can be accepted as present form.

R= We appreciate your time and support to improve the present manuscript.

Round 2

Reviewer 1 Report

Thank you for responding to all of the questions. The manuscript has improved, however I still feel that the overall analysis is very simplistic and largely descriptive. There is no modelling or any kind that shows adjustments for the demographic data collected. Most of the discussions are based around females being more health conscious than males. Please note that once BMI is in Z-scores you can also do whole group analysis without separating by age an sex (as these are accounted for in the calculation of Z-score)

Results:

Table 1: Why is water not included in this table, or is it part of SF beverages? If it is, it should be separated. It makes sense to include water because then Figure 1 shows the % of SSB in relation to total fluid intake. 

Lines 176-179: I do not know what the authors are trying to say here...Why or how was it decided that 26 grams of sugar was considered high? How did you determine these cut offs to separate the groups- it is not really acceptable to use quantiles as it can really skew the interpretation of findings. Is there a meaningful cut-off you can use- something that is part of your countries guidelines. If your cut-offs are in fact informed by the literature then you should mention this as part of your analysis/methods.

Why not just show a correlation between grams of sugar and age as continuous variables?

Figure 1 itself: Daily sugar consumption (%)-  What is this a % of? Daily total beverage intake? This is what I am assuming so it should be clearer- perhaps a footnote explaining how it was calculated. 

Why isn't water in the any of the tables/figures? I would think it is important to include? or is it part of SF-beverages. if i is it should be separated

Lines 186- 191: again this is a little difficult to understand in the way it is written. Perhaps something like  sugar intake, energy intake and SS-B were all negatively correlated with BMI Z-score in girls with normal weight. So you are saying that among NBM girls as their weight increases sugar intake decreases?  Again I am wondering why you are separating them into so many groups for correlation analysis. Why not whole group analysis or just separate boys and girls. The correlations are very weak. So while some are statistically significant it is not biologically meaningful. Except for 12 year old boys where as weight increases their consumption of SF beverages decrease. You would think the correlations for other drinks would be in the opposite direction

there is only one limitation discussed. The use of a semiquantitative beverage questionnaire is also a limitation because these types of tools generally over estimate intake. Perhaps diaries should have been kept. The use of arbitrary cut-offs to mark high sugar consumption is limiting, as is the simple analysis plan that is mostly descriptive.

Whilst I appreciate the efforts to improve the manuscript I do believe a lot more could be done here to improve analysis so something more novel is presented.

Author Response

We truly appreciate your contributions. The changes requested can be identified in the new version of our manuscript (ijerph-1687756-R2) while a point-by-point response to your requests is described below:

Thank you for responding to all of the questions. The manuscript has improved, however I still feel that the overall analysis is very simplistic and largely descriptive. There is no modelling or any kind that shows adjustments for the demographic data collected. Most of the discussions are based around females being more health conscious than males. Please note that once BMI is in Z-scores you can also do whole group analysis without separating by age an sex (as these are accounted for in the calculation of Z-score)

R= We appreciate all your observations to improve the quality of our manuscript. In order to further clarify the results, the findings reported in the correlation analysis have made simpler and show possible associations between the types of beverages analyzed, and the differences between genders are discussed with relevant reservations, more specific comments about it are described in the next answers.

Results:

Table 1: Why is water not included in this table, or is it part of SF beverages? If it is, it should be separated. It makes sense to include water because then Figure 1 shows the % of SSB in relation to total fluid intake. 

R= Effectively, water is part of sugar-free beverages, and according your suggestion, we have added to the Table the intake of water, coinciding with you that it could give more sense to Figure 1, thank you.

Lines 176-179: I do not know what the authors are trying to say here...Why or how was it decided that 26 grams of sugar was considered high? How did you determine these cut offs to separate the groups- it is not really acceptable to use quantiles as it can really skew the interpretation of findings. Is there a meaningful cut-off you can use- something that is part of your countries guidelines. If your cut-offs are in fact informed by the literature then you should mention this as part of your analysis/methods.

R= It was decided 25 g of sugar intake due is the WHO recommendation. To avoid any misunderstood of the used cut-offs, we have added the respective reference and the following statement in the discussion section “The WHO recommends an intake of free sugars roughly 25 g (below 5% of total energy intake) in an adult with normal BMI [47]. Our results suggest that most children between 9 and 12 years consume more sugar from beverages than the WHO guidelines recommendations in adults, which is alarming due the 5% of total energy intake in children is still fewer 25 g/day, also, sugar intake of solid foods are not considered in this research”.

Why not just show a correlation between grams of sugar and age as continuous variables?

R= In fact, WHO AnthroPlus Software uses age in days, which allows us to get closer to the continuity of this variable. Moreover, to further clarify the findings, the rows of age strata were removed in to generate a simpler Table 3. Thanks for your observation.

Figure 1 itself: Daily sugar consumption (%)-  What is this a % of? Daily total beverage intake? This is what I am assuming so it should be clearer- perhaps a footnote explaining how it was calculated. 

R= according to your observation, we have detected that we made a mistake in the y axis name so we have corrected it, it refers to the percentage of children,  thanks for your observation.

Why isn't water in the any of the tables/figures? I would think it is important to include? or is it part of SF-beverages. if it is it should be separated

R= According to your first comment and this one, in Table 2, we have added the intake of water according, age, sex and BMI Z-score, thank you.

Lines 186- 191: again this is a little difficult to understand in the way it is written. Perhaps something like  sugar intake, energy intake and SS-B were all negatively correlated with BMI Z-score in girls with normal weight. So you are saying that among NBM girls as their weight increases sugar intake decreases?  Again I am wondering why you are separating them into so many groups for correlation analysis. Why not whole group analysis or just separate boys and girls. The correlations are very weak. So while some are statistically significant it is not biologically meaningful. Except for 12 year old boys where as weight increases their consumption of SF beverages decrease. You would think the correlations for other drinks would be in the opposite direction

R= According to your observation, the rows of age strata were removed in to generate a simpler Table 3. However, the paragraph have been rewritten to describe the results in a better way, resulting in the following statement: “Interestingly, BMI Z-score in girls and boys didn’t present significant correlations with SSB intake, only girls with excess body mass showed a positive correlation with sugar-free beverages consumption. In contrast, dairy and sugar-free beverages intake in the total of boys and boys with excess body mass were negatively correlated with the BMI Z-score.”.

Although it is true that the consumption of one or another type of beverage contributes to the pathogenesis of obesity and its consequences, there are other determinants such as the caloric intake of other types of food (global energy consumption) or levels of physical activity that were not considered in the present study. These limitations have already been stated in the Discussion section.

there is only one limitation discussed. The use of a semiquantitative beverage questionnaire is also a limitation because these types of tools generally over estimate intake. Perhaps diaries should have been kept. The use of arbitrary cut-offs to mark high sugar consumption is limiting, as is the simple analysis plan that is mostly descriptive.

R= According to your observation, we have discussed more limitations of the study, including the use of semiquantitative beverage questionnaire,  the lack of information about sugar intake from solid foods, and that is not possible to determine causality between BMI and the factors evaluates due this is only a cross-sectional study. All limitations are included at the final of the Discussion section, thank you.

Whilst I appreciate the efforts to improve the manuscript I do believe a lot more could be done here to improve analysis so something more novel is presented.

R= We hope that attention to the points that have been discussed in all the manuscript sections culminated in a more important research work. The authors are aware of the scope of the study and one of the objectives of the research group is to set a precedent  for a future longitudinal study including a nutritional program and physical activity intervention in children from the Mexico-USA border.

Sincerely

Dr. Marco Antonio Hernández-Lepe, Corresponding author

This manuscript is a resubmission of an earlier submission. The following is a list of the peer review reports and author responses from that submission.

Round 1

Reviewer 1 Report

The article presented is of great interest and relevance. There is a worldwide obesity problem with the consumption of sugar-sweetened beverages that is particularly dramatic in the case of children. 

Below are a number of aspects that could improve the work presented:
- In the methodology, the authors should first specify the type of study and the aspects related to the design to finish with the ethical aspects.
- It would be advisable for the authors to elaborate on both the limitations and the possible lines of future work based on the research carried out.

Author Response

Reviewer 1

We truly appreciate your contributions. The changes requested can be identified in the new version of our manuscript (ijerph-1639652-R1) while a point-by-point response to your requests is described below:

The article presented is of great interest and relevance. There is a worldwide obesity problem with the consumption of sugar-sweetened beverages that is particularly dramatic in the case of children. 

Below are a number of aspects that could improve the work presented:
- In the methodology, the authors should
first specify the type of study and the aspects related to the design to finish with the ethical aspects.

R= We specified the type of study and the Materials and Methods section has been reorganized according to your recommendation, thank you.

- It would be advisable for the authors to elaborate on both the limitations and the possible lines of future work based on the research carried out.

R= According to your observation, we added information in the Discussion section, including the following paragraphs:

“The principal limitation of this study is the lack of physical activity information in the childhood, moreover, public schools in Mexico offers only one hour of physical education lessons per week, with an effective time of 39±0.7 min, of which only 41% (15 min) is intended for moderate-to-vigorous intensity activities such as soccer or basketball. In this sense, physical activity in the primary school does not correspond to the minimum recommendations for a healthy life and additional strategies are required for the promotion and achievement of well-being.

According to the aforementioned, there is an urgency to develop strategies focused to counteract the high rate of childhood overweight and obesity. Future strategies should consider a multidimensional model that includes, among other determinants: healthy food options in school stores, developing food preparation skills, encourage physical activity beyond school (by offering different types of exercises or sports to increase the possibility that children will be motivated to move by finding something they like) and consider to evaluate more specific parameters of body composition, such as waist girth, hip girth, and body fat percentage by bioelectrical impedance analysis, anthropometry or densitometric methods. We consider that the adaptation of protocols that contemplate a holistic view of this problem and the consideration of specific markers, will culminate in more effective strategies that could help them adopt a healthy lifestyle last throughout life”.

According to all your observations, the article has been improved substantially. We really appreciate your contributions.

Reviewer 2 Report

The article entitled "Association between body mass index with sugar-sweetened and dairy beverages consumption in children from the Mexico-USA border" presents data on the study conducted on the total of 720 elementary school children. The significant number of children participated in the study undoubtedly raises the importance of the presented research.

General comments:

There is no information how the BMI was calculated and what are the interpretation criteria for BMI,

As the authors admitted in the Discussion section the information about physical activity of children is missing in the study and therefore I would suggest to chceck if it is possible to add at least information on physical activity in a frame of school acitvities (how many hours, what is the frequence of taking part in such activities etc.).

Specific comments:

Line 47 - comma should be used between the number of references

Line 81 - Reference 19 should be placed at the end of the sentence, next to the ref. 20

Line 169 - Reference 23 should be placed at the end of the sentence

Figure 2 - add more precise description to this figure, what do the colors stands for?

Table 3 - In column BMI, there is no information about BMI, please revise this column.

Line 195 - Reference 25 should be placed at the end of the sentence together with ref. 26 and 27

Line 205 - Reference 31 should be placed at the end of the sentence together with ref 32. The same stuation is with reference 30 and 40 (lines 217 and 224).

Line 246 - When citing Della Corte et al., in brackets there should be year of publication placed and not the reference number, which should be at the end of the sentence.

Line 346-347 - Reference 29 should have last accesion date provided.

Author Response

The article entitled "Association between body mass index with sugar-sweetened and dairy beverages consumption in children from the Mexico-USA border" presents data on the study conducted on the total of 720 elementary school children. The significant number of children participated in the study undoubtedly raises the importance of the presented research.

General comments:

There is no information how the BMI was calculated and what are the interpretation criteria for BMI,

R= According to your observation, we added information in Section 2.4, including the following statement: “The BMI was calculated by taking the participants weight, in kilograms, divided by their height, in meters squared (BMI= weight (kg)/ [height (m)]2. The BMI was converted into age and sex specific BMI percentiles, defining normal body mass as above the 15th and below the 85th percentile and excess of body mass above the 85th percentile, in order to classify children’s BMI according to the World Health Organization growth reference for school-aged children and adolescents”.

As the authors admitted in the Discussion section the information about physical activity of children is missing in the study and therefore I would suggest to check if it is possible to add at least information on physical activity in a frame of school activities (how many hours, what is the frequence of taking part in such activities etc.).

R= We considered your suggestion. The following statement has been added to the Discussion section: “The principal limitation of this study is the lack of physical activity information in the childhood, moreover, public schools in Mexico offers only one hour of physical education lessons per week, with an effective time of 39±0.7 min, of which only 41% (15 min) is intended for moderate-to-vigorous intensity activities such as soccer or basketball. In this sense, physical activity in the primary school does not correspond to the minimum recommendations for a healthy life and additional strategies are required for the promotion and achievement of well-being”.

Specific comments:

Line 47 - comma should be used between the number of references

R= The manuscript had a dot instead a comma in references 3,4 and 5,6. Thanks for your observation.

Line 81 - Reference 19 should be placed at the end of the sentence, next to the ref. 20

R= We have changed the requested reference.

Line 169 - Reference 23 should be placed at the end of the sentence

R= We have changed the requested reference.

Figure 2 - add more precise description to this figure, what do the colors stands for?

R= We have improved Figure 2 and added caption description of the colors meaning in the manuscript, thank you.

Table 3 - In column BMI, there is no information about BMI, please revise this column.

R= We have added BMI units in the column and detailed that it is divided by gender, thanks for your observation.

Line 195 - Reference 25 should be placed at the end of the sentence together with ref. 26 and 27

R= We have changed the requested reference.

Line 205 - Reference 31 should be placed at the end of the sentence together with ref 32. The same stuation is with reference 30 and 40 (lines 217 and 224).

R= We have changed the requested references.

Line 246 - When citing Della Corte et al., in brackets there should be year of publication placed and not the reference number, which should be at the end of the sentence.

R= We have changed the requested reference, in order that the Editor confirms that it is correctly cited.

Line 346-347 - Reference 29 should have last accesion date provided.

R= Last access has been added in reference 29 .

According to your observations, the article has been improved substantially. We really appreciate your contributions.

Reviewer 3 Report

The paper in question is correctly done, although it is very simple and does not seem to find a strong relationship between SSB and BMI, possibly because the BMI depends on a wide variety of factors.

Regarding the realization of the paper, the main improvement needed is Figure 2, as it is not understood and is essentially a black mark.

Author Response

Reviewer 3

We truly appreciate your contributions. The changes requested can be identified in the new version of our manuscript (ijerph-1639652-R1) while a point-by-point response to your requests is described below:

The paper in question is correctly done, although it is very simple and does not seem to find a strong relationship between SSB and BMI, possibly because the BMI depends on a wide variety of factors.

R= Your observation was considered, so we have added information about the possible lines of future work focused on counteract factors associated with excess body mass in children, including the following paragraph in the Discussion section: “There is an urgency to develop strategies focused to counteract the high rate of childhood overweight and obesity. Future strategies should consider a multidimensional model that includes, among other determinants: healthy food options in school stores, developing food preparation skills, encourage physical activity beyond school (by offering different types of exercises or sports to increase the possibility that children will be motivated to move by finding something they like) and consider to evaluate more specific parameters of body composition, such as waist girth, hip girth, and body fat percentage by bioelectrical impedance analysis, anthropometry or densitometric methods. We consider that the adaptation of protocols that contemplate a holistic view of this problem and the consideration of specific markers, will culminate in more effective strategies that could help them adopt a healthy lifestyle last throughout life”.

Regarding the realization of the paper, the main improvement needed is Figure 2, as it is not understood and is essentially a black mark.

R= We have improved Figure 2. According to your observations, the article has been improved substantially. We really appreciate your contributions.